# Impact of Physician Characteristics on Late-Onset Sepsis (LOS) Evaluation in the NICU

**DOI:** 10.3390/healthcare12080845

**Published:** 2024-04-17

**Authors:** Harshkumar R. Patel, Blaine Traylor, Mohamed Farooq Ahamed, Ginger Darling, Albert Botchway, Beau J. Batton, Venkata Sasidhar Majjiga

**Affiliations:** 1Department of Pediatrics, SIU School of Medicine, Springfield, IL 62794, USA; hpatel77@siumed.edu (H.R.P.); gdarling@siumed.edu (G.D.); bbatton@siumed.edu (B.J.B.); 2SIU School of Medicine, Springfield, IL 62794, USA; btraylor61@siumed.edu; 3Center for Clinical Research, SIU School of Medicine, Springfield, IL 62702, USA; abotchway38@siumed.edu

**Keywords:** late-onset sepsis, neonatal sepsis, late-onset sepsis screening, physician characteristics, physician bias

## Abstract

The threshold for a late-onset sepsis (LOS) evaluation varies considerably across NICUs. This unexplained variability is probably related in part to physician bias regarding when sepsis should be “ruled out”. The aim of this study is to determine if physician characteristics (race, gender, immigration status, years of experience and academic rank) effect LOS evaluation in the NICU. This study includes a retrospective chart review of all Level III NICU infants who had a LOS evaluation over 54 months. Physician characteristics were compared between positive and negative blood culture groups and whether CBC and CRP were obtained at LOS evaluations. There were 341 LOS evaluations performed during the study period. Two patients were excluded due to a contaminant. Patients in this study had a birth weight of [median (Q1, Q3)]+ 992 (720, 1820) grams and birth gestation of [median (Q1, Q3)] 27^6/7^ (25^2/7^, 33^0/7^) weeks. There are 10 neonatologists in the group, 5/10 being female and 6/10 being immigrant physicians. Experienced physicians were more likely to obtain a CBC at the time of LOS evaluation. Physician characteristics of race, gender and immigration status impacted whether to include a CRP as part of a LOS evaluation but otherwise did not influence LOS evaluation, including the likelihood of bacteremia.

## 1. Introduction

Despite advances in neonatal care, late-onset sepsis (LOS) remains a significant cause of morbidity and mortality among neonates [1,2,3,4,5]. Surviving infants affected by LOS are at an increased risk of adverse neurodevelopmental outcomes including hearing loss, cerebral palsy, visual impairment, and cognitive delays [2,4,6,7,8,9]. Timely and accurate evaluations of neonates with potential sepsis in the neonatal intensive care units (NICU) are crucial for improving outcomes.

The incidence of LOS ranges from 2% to 5% of all NICU admissions and can be as high as 10% to 30% in very low birth weight (VLBW) infants [3,10]. LOS is defined as a positive blood culture in sepsis evaluation conducted more than 72 h after birth. Early recognition and diagnosis of LOS remain challenging, as the signs and symptoms of sepsis in neonates can vary and often can be subtle or nonspecific, despite their potentially serious consequences [11,12]. Presentation of LOS in neonates includes, but is not limited to, respiratory signs (apnea, desaturations, increase in work of breathing), lethargy, tachycardia, feeding intolerance, and temperature instability. The severity of the illness can range from mild signs to critical illness involving severe organ dysfunction and potential multiorgan failure [11,13]. A LOS evaluation can be influenced by patient characteristics and physician characteristics in addition to patients’ symptoms. The effect of physician characteristics on clinical decision making has not been studied in neonates.

Approximately 67 to 83% of LOS evaluations are associated with a negative blood culture [10,14]. It is not uncommon for an infant to undergo evaluation for suspected sepsis and receive empiric antibiotics for 36 to 48 h while sepsis is being “ruled out”. There is evidence that even brief exposure to antibiotics in the newborn period can alter intestinal microbiota with potentially long-term health implications, with an increase in risk of necrotizing enterocolitis and antibiotic resistance [15,16,17]. Implications of exposure to antibiotics in neonatal period include wheezing and allergies in later childhood, obesity, and inflammatory bowel diseases [18,19,20,21].

Late-onset sepsis usually involves diverse pathogens from various sites (blood, skin and soft tissue, GI tract, urinary tract, and lung) compared to early-onset sepsis (EOS) from most prevalent bacteria (*Escherichia coli*, Group B *Streptococcus* (GBS), or *Listeria*) [22]. The guidelines for the management of EOS have consistently demonstrated efficacy in reducing the severity of morbidity of EOS [23]. Conversely, similar evidence-based guidelines for the management of LOS in neonatal intensive care units are lacking, likely due to larger regional variability in pathogens causing LOS. A physician survey in 2002 showed that CBC is ordered in 99% and CRP in 32% of all LOS evaluations, with similar trends seen in an EOS evaluation survey in 2017 [24,25]. The role of laboratory tests like C-reactive protein (CRP), complete blood count (CBC), interleukins (IL), and procalcitonin for the diagnosis of LOS is questionable due to the poor positive predictive value of these tests. This emphasizes the clinical judgment of clinicians in making a diagnosis of LOS [26,27,28,29].

Currently, there is no consensus on a standardized criterion for LOS evaluation, and the role of inflammatory markers in LOS evaluation is questionable. Therefore, LOS evaluations heavily rely on the physician’s discretion and can vary depending on the interpretation of the symptoms, leading to subjective bias and potential variations in diagnostic practice. Physician characteristics, such as experience, expertise, and clinical judgment, may play a critical role in LOS diagnosis and management. However, the impact of these physician-specific factors on LOS evaluation remains poorly understood. Previous research has identified considerable practice variation among neonatologists in managing suspected neonatal sepsis in the NICU [30]. Nevertheless, physician characteristics have not been explored, specifically in the context of LOS evaluation in the NICUs. This study was conducted to investigate the impact of physician characteristics on evaluations for LOS in the NICU and to shed light on potential variations in diagnostic practices and management decisions.

We believe that understanding the impact of physician characteristics on evaluations for late-onset sepsis in the NICU is crucial for optimizing clinical decision making and improving patient outcomes. Our study aims to bridge the existing knowledge gap by exploring the association between physician characteristics and the diagnostic process for LOS in the NICU setting.

## 2. Methods

### 2.1. Study Design

This retrospective chart review aimed to investigate the impact of physician characteristics (race, gender, immigration status, years of experience and academic rank) on evaluations for late-onset sepsis (LOS) in a Level III NICU. Given the potential variation in the medical practices among physicians from different backgrounds and places of physician training, we included the immigration status to explore its impact on the diagnosis of LOS. Each physician characteristic was compared between three groups: 1. Blood culture positive and negative group; 2. CRP carried out/not carried out at LOS evaluation, and 3. CBC carried out/not carried out at LOS evaluation.

### 2.2. Study Population

The study included all NICU patients with suspected LOS who had a blood culture obtained at more than 72 h of life in a 56-bed Level III NICU (with more than 800 annual admissions and with an average daily census of 41). As policy in our unit, a blood culture is carried out in all suspected cases of LOS prior to initiation of antibiotics. Even though we have midlevel trainees and providers in our unit, the final decision to do a LOS evaluation is carried out by a neonatologist. We have 24/7 an in-house neonatologist in our NICU. Data were collected from March 2018 to September 2022.

Infants who had received any antibiotics prior to LOS evaluation were excluded from the study. LOS was defined as a positive blood culture at more than 72 h of age that was not considered a contaminant. A blood culture was determined as a contaminant when it grew skin commensal bacteria (coagulase-negative Staphylococcus being the most common) and based on a physician’s clinical decision to stop antibiotics in less than 5 days. Contaminants were excluded from the study, as we did not want them to affect our analysis of correlation of physician characteristics and a true positive blood culture. For neonates with multiple LOS evaluations, each sepsis evaluation was considered a separate event for the purpose of this study.

The study protocol was approved by the local Institutional Review Board, with the waiver of consent due to the retrospective nature of this study.

### 2.3. Data Collection

Physician characteristics were collected, including years of experience, gender, race, immigration status, and academic rank. To ensure confidentiality, physician names and identities were not disclosed. Study investigator HP (a resident), along with BT (a medical student), collected the physicians’ information. Other physicians involved in the study were blinded to physician identities. The determination of which physician did LOS evaluation was based on physician note in regard to LOS evaluation, and in the absence of a physician note, a nursing documentation of communications with the physician and on-call roster was used to determine the physician who carried out the LOS evaluation.

For each case, the following variables were collected: patient characteristics (gender, weight, and gestational age at birth), signs and symptoms of sepsis, blood culture results, laboratory parameters CBC, CRP, and duration of antibiotic use.

### 2.4. Statistical Analysis

Statistical analysis was performed using the Stata software version 14 (Collage Station, TX, USA). A comparison was made between the positive and negative blood culture groups in LOS evaluations for each physician characteristic. We also examined whether physician characteristics were associated with the decision to perform CBC and CRP tests during LOS evaluations.

The Fisher exact test was used to compare the positive blood culture group with the negative blood culture group for categorical variables, while the Mann–Whitney test was used for continuous variables. Continuous variables were reported as medians, and categorical variables were reported as percentages. Statistical significance was defined as a *p*-value less than 0.05.

### 2.5. Patient Participation Statement

Patients or their legal guardians were not involved in the design, recruitment, or conducting of the study, as this is a retrospective study.

## 3. Results

During the 56-month study from March 2018 to September 2022, there were 3460 admissions to the NICU, of whom 341 (10%) underwent a LOS evaluation, of which 52 (15.2% of LOS evaluations; 1.5% of all NICU admissions) had a positive blood culture. Two patients with positive cultures were excluded, as they were determined to be contaminants.

The median (interquartile range) values for birth weight (BW) and gestational age (GA) at birth for patients who underwent a LOS evaluation were 992 (720, 1820) grams and 27^6/7^ (25^2/7^, 33^0/7^) weeks, respectively. The median (interquartile range) weight at LOS evaluation was 1762.5 (1000, 2865) grams. The median (interquartile range) post-menstrual age (PMA) at LOS evaluation was 33^4/7^ (29^0/7^, 38^3/7^) weeks, and the median (interquartile range) age was 22 (10, 45) days (Table 1).

There were 10 neonatologists in the group, of whom half were female. Six physicians were immigrants from four different countries. Four physicians were from the USA, three were Indian, one Haitian, one Colombian, and one was Lebanese (5/10 female, 6/10 immigrants, and 3/10 were white physicians). The academic ranks for the eight neonatologists at the beginning of the study were as follows: four assistant professors and four associate professors. By the end of the study, the neonatologist group increased to ten, and the distribution had changed to three assistant professors, four associate professors, and three professors. The range of years of experience among the physicians was from 1 to 18 years.

The nine common indications for LOS evaluation were abdominal distention (9%), increase in respiratory support (25%), significant respiratory distress/apnea needing intubation (14%), tachycardia (3%), fever (3%), lethargy (6%), recurrent apnea/bradycardia/desaturations (25%), bloody stools (5%), and soft-tissue infections (10%). Other indications were less common, and some patients had multiple indications.

Among the 339 total cases included in the study, 324 (95.6%) had a CBC performed during LOS evaluation (Table 2). There were no significant differences in obtaining CBC as part of LOS evaluations based on physician race (*p* = 0.540), gender (*p* = 0.999), immigration status (*p* = 0.078), and academic rank (*p* = 0.468). However, there was a statistically significant association between years of experience and CBC ordering at LOS evaluation (*p* = 0.045) (median experience when CBC was obtained was 10 years compared to 6 years when CBC was not obtained).

Out of the 339 cases, 174 (51.3%) had a CRP test performed during the LOS evaluation. There was a significant association between physician race and CRP testing (*p* = 0.006), with a higher proportion of non-white physicians ordering CRP tests. Gender also showed a significant association with CRP ordering (*p* < 0.001), as more male physicians ordered CRP tests compared to their female counterparts. Immigration status was found to be significantly associated with CRP testing (*p* < 0.001), with a higher proportion of immigrant physicians ordering CRP tests. No significant association was found for years of experience and academic rank in regard to ordering CRP at LOS evaluation (*p* > 0.05) (Table 3).

Out of the 339 cases, 52 (15.3%) had a positive blood culture. Two patients were excluded, as the blood cultures were determined as contaminants (which grew coagulase-negative Staphylococcus and antibiotics were given for less than 5 days by treating physician). The following organisms grew in the positive cultures: 22 coagulase-negative Staphylococcus (44%), 7 Staphylococcus aureus (14%), 4 Escherichia coli (8%), 1 Pseudomonas aeruginosa (2%), 6 Klebsiella pneumoniae (12%), 2 Enterococcus faecalis (4%), 6 Group B Streptococcus (12%) and 2 methicillin-resistant Staphylococcus aureus (MRSA) (4%). There were no significant differences seen in the likelihood of a positive blood culture based on physician race (*p* = 0.728), gender (*p* = 0.547), years of experience (*p* = 0.914), immigration status (*p* = 0.999), or academic rank (*p* = 0.764) (Table 4).

## 4. Discussion

Late-onset sepsis remains a significant concern in NICUs despite advancements in neonatal care, with potential adverse neurodevelopmental outcomes [2,4,6,7,8,9]. The lack of guidelines for LOS evaluation puts the onus on physicians to make the decision and increase the potential variation in practice. There is evidence of racial and ethnic disparity in quality of care in the NICU [31] but there is no available literature looking at physician characteristics at LOS evaluation.

A study in adult patients carried out by Reid et al. showed that few characteristics of individual physicians were associated with higher performance on measures of quality, and associations were small in magnitude [32]. Tussing and Worrowycz found in their study in 1993 that physician characteristics like experience, US or foreign medical graduation, gender, board certification and professional appointment affect Cesarean section rates and rates of dystocia and fetal distress [33]. On the other hand, a physician survey on implementation of practice guidelines has shown a positive impact on their practice in recent graduates, women and minorities [34].

This study aimed to investigate the impact of physician characteristics on evaluations for LOS and to shed light on diagnostic practice variations. This study found that physician characteristics had a varying influence on the diagnostic process for LOS. One important aspect examined in this study was the use of diagnostic tools, specifically the ordering of CBC and CRP tests at LOS evaluations. It is reported in the literature that 99% and 95% of physicians obtain CBC as part of EOS and LOS evaluations, respectively [24,25]. This study showed that years of experience significantly influenced the ordering of CBC tests, with more CBC ordered by physicians who had been in practice for a longer time. This finding needs to be further substantiated in a larger study and with physician interviews to evaluate factors that influence their decision making. One likely reason for this could be the emerging evidence questioning the role of CBC in LOS evaluations. The analysis also revealed a significant association between physician characteristics and the ordering of CRP tests during LOS evaluation. Non-white physicians, male physicians, and immigrant physicians were more likely to order CRP tests compared to their counterparts. These findings highlight variations in diagnostic practices based on physician characteristics, which may have implications for resource use and patient management. With recent publications questioning the utility of CRP in LOS evaluation, there is scope for unifying physician practice with national guidelines for screening LOS [29,35].

This study did not find significant differences in the association between any of the physician characteristics and the likelihood of a positive blood culture in LOS evaluations. This suggests that physician characteristics may influence the use of diagnostic tools, such as CBC and CRP tests, at LOS evaluations but do not strongly affect the identification of infants who are at risk for late-onset bacteremia, again questioning the utility of CBC and CRP in routine LOS evaluation.

This study was not intended to look at patient factors affecting physician decision making for LOS evaluation. The following are the limitations of this study: (1) This study was retrospective, relying on chart reviews and collected data. Prospective studies with larger sample sizes and more diverse populations are needed to validate and generalize these findings. (2) This study was conducted in a single NICU with only 10 neonatologists, which may limit the generalizability of the results to other settings. (3) A physician’s intention in ordering CBC and CRP was not evaluated, as this is a retrospective study. A prospective study with a physician questionnaire is needed to address this limitation. (4) Each physician’s variation in LOS evaluation was not analyzed, as each LOS assessment was treated as an independent event. This could be addressed with a large dataset of physicians, so that individual physician LOS evaluations can be compared at different periods (based on years of experience or academic rank). (5) The impact of lawsuits on physician decision making has not been evaluated in this study, as we felt physicians would be uncomfortable answering questions related to lawsuits. This is a contentious topic to be addressed in research, and an anonymous survey needs to be conducted to address this issue. A multi-center study involving NICUs with distinctive characteristics would provide a more comprehensive understanding of the impact of physician characteristics on LOS evaluations.

Addressing variation in LOS evaluation between physicians can help with the antibiotic stewardship effort in the NICU. Moving forward, it is essential to explore innovative approaches to enhance the diagnostic process and improve patient outcomes in late-onset sepsis evaluations. Earlier detection of pathogens by rapid nucleic acid-based identification tests, like peptide nucleic acid–fluorescence in situ hybridization molecular stains (PNA-FISH) and PCR-based tests by analysis of their DNA by polymerase chain reaction (PCR), will yield faster results with higher sensitivity with potential for decreasing variation in physician LOS evaluations [36,37]. Modern technologies, such as artificial intelligence (AI), hold promise in this regard. AI algorithms can analyze large volumes of patient data, including clinical signs, laboratory results, and imaging findings, to assist physicians in early and accurate diagnosis [38]. By integrating AI tools into the diagnostic workflow, healthcare providers can benefit from improved efficiency and decision support, leading to better detection and management of late-onset sepsis while decreasing the influence of physician characteristics [39].

## 5. Conclusions

Our study showed that physician characteristics, such as years of experience, gender, race, and immigration status, influence the diagnostic process for late-onset sepsis in the NICU. However, no significant associations were found between physician characteristics and the presence of a positive blood culture. These findings emphasize the complexity of LOS evaluations and the need for standardized protocols to ensure consistent and accurate diagnoses.

## Figures and Tables

**Table 1 healthcare-12-00845-t001:** Patient characteristics at LOS evaluation.

Patient Characteristics	Median (Q1, Q3)
Birth weight	992 (720, 1820)
Gestational age at birth	27^6/7^ (25^2/7^, 33)
Weight at LOS evaluation	1762.5 (1000, 2865)
Post-menstrual age at LOS evaluation	33^4/7^ (29^0/7^–38^3/7^)
Day of life at LOS evaluation	22.0 (10.0, 45.0)

**Table 2 healthcare-12-00845-t002:** Physician characteristics and association with obtaining a CBC at time of LOS evaluation.

Physician Characteristics		Yes (*n* = 324)	No (*n* = 15)	Total *(N* = 339)	*p*-Value
**Race**	White	79 (94.0%)	5 (6.0%)	84	0.540
	Non-white	245 (96.1%)	10 (3.9%)	255	
**Gender**	Male	170 (95.5%)	8 (4.5%)	178	0.999
	Female	154 (95.7%)	7 (4.3%)	161	
**Years of experience**	Median (Q1, Q3)	10 (4, 13)	6 (2, 11)	10 (4, 13)	0.045 *
**Immigration status**	Non-immigrant	93 (92.1%)	8 (7.9%)	101	0.078
	Immigrant	231 (97.1%)	7 (2.9%)	238	
**Academic rank**	Assistant Professor	123 (93.9%)	8 (6.1%)	131	0.468
	Associate Professor	188 (96.4%)	7 (3.6%)	195	
	Professor	13 (100%)	0 (0%)	13	

* *p* < 0.05 was considered as statistically significant.

**Table 3 healthcare-12-00845-t003:** Physician characteristics and association with obtaining a CRP at time of LOS evaluation.

Physician Characteristics		Yes (*n* = 174)	No (*n* = 165)	Total (*N* = 339)	*p*-Value
**Race**	White	32 (38.0%)	52 (62.0%)	84	0.006 *
	Non-white	142 (55.7%)	113 (44.3%)	255	
**Gender**	Male	109 (61.2%)	69 (38.8%)	178	<0.001 *
	Female	65 (40.3%)	96 (59.7%)	161	
**Years of experience**	Median (Q1, Q3)	10 (4, 13)	9 (5, 14)	10 (4, 13)	0.797
**Immigration status**	Non-immigrant	36 (35.6%)	65 (64.4%)	101	<0.001 *
	Immigrant	138 (58.0%)	100 (42.0%)	238	
**Academic rank**	Assistant Professor	61 (46.5%)	70 (53.5%)	131	0.405
	Associate Professor	106 (54.3%)	89 (45.7%)	195	
	Professor	7 (53.8%)	6 (46.2%)	13	

* *p* < 0.05 was considered as statistically significant.

**Table 4 healthcare-12-00845-t004:** Physician characteristics and association with a positive blood culture with LOS evaluation.

Physician Characteristics		Positive (*n* = 52)	Negative (*n* = 287)	Total (*N* = 339)	*p*-Value
**Race**	White	14 (16.6%)	70 (83.4%)	84	0.728
	Non-white	38 (14.9%)	217 (85.1%)	255	
**Gender**	Male	25 (14.0%)	153 (86%)	178	0.547
	Female	27 (16.8%)	134 (83.2%)	161	
**Years of experience**	Median (Q1, Q3)	9 (5.5, 12)	10 (4, 13)	10 (4, 13)	0.914
**Immigration status**	Non-immigrant	15 (14.8%)	86 (85.2%)	101	0.999
	Immigrant	37 (15.5%)	201 (84.5%)	238	
**Academic rank**	Assistant Professor	19 (14.5%)	112 (85.5%)	131	0.764
	Associate Professor	32 (16.4%)	163 (83.6%)	195	
	Professor	1 (7.7%)	12 (92.3%)	13	

## Data Availability

The original contributions presented in the study are included in the article, further inquiries can be directed to the corresponding author. If warranted, raw deidentified data supporting the conclusions of this article may be made available by the authors on request.

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
