# Peer review of "Impact of Physician Characteristics on Late-Onset Sepsis (LOS) Evaluation in the NICU"

_healthcare, 2024, doi:10.3390/healthcare12080845_

Round 1

Reviewer 1 Report

Comments and Suggestions for Authors

I could not understand what this result contributes for clinical practice. For example, there is the problem of antibiotics being routinely prescribed for viral infections. Not only there is no advantage, but also there is even a disadvantage. In the same way, there is little significance in testing CRP and CBC as indicators to determine LOS (which the author also describes in the introduction). I recommend that you add in the introduction how much CRP and CBC are actually tested in clinical practice and whether there are differences in the judgment of such tests among physicians.  

Another comment is that you could not evaluate that whether or not culture testing is used as an indicator to diagnose LOS affects physician characteristics. Even if the results of the test were Negative (or even Positive), the physician would have done the test, which would indicate that the physician decided that a culture test was necessary to diagnose LOS. Although not possible in this retrospective study, it may be important for further research to ascertain the percentage of culture tests performed for the purpose of diagnosing LOS.

Author Response

Thank you for your time to review this manuscript and your valuable comments. Please see our response uploaded.

Reviewer 2 Report

Comments and Suggestions for Authors

This paper compares lab ordering practices among a group of ten neonatologists when evaluating patients for sepsis after 3 days of life.

Title: Does not specify NICU setting, which needs to be added. I would replace “at” with “on”

Abstract: line 17: add something like “were obtained” after “whether CBC and CRP”

Final sentence should include what characteristics were affected.

Introduction: The authors do not use acronyms consistently after they are defined. For example, they define LOS on line 29, then write it out completely on line 37 and line 63.
Line 52: replace “;” with “,”

Line 57: remove capital E in the middle of the sentence.

Line 59:  Managing something does not reduce its risk, it minimizes the severity of its morbidity.

Line 60: What is “EOS”? Acronym never defined.

Methods: Line 101 says you included “commensals” as contaminants. However, isn’t coagulase-negative staph, which was your #1 “positive” organism (line 181), usually considered such a contaminant? This really needs to be discussed / addressed!

Results:

Lines 146 and 149: was it 8 neonatologists, or 10? If the group grew during the study period, that is OK but you need to tell the readers.

Lines 154-157: These add to 100%, and there are more. Please tell the readers that some patients had multiple “reasons” – maybe a better word is “indications” – for a LOS workup.

The authors have missed a big opportunity to look for correlations between the results of the CBC and CRP tests and the culture results.

Discussion: Line 197: Since neonatologists are adults, use of the word “adults” here does not distinguish the two studies. Say more, like maybe “adult patients”?

Paragraph beginning on line 205: Neonatologists who have been in practice longer are considered here to have more “years of experience” and that is true. However, the authors might consider, as an explanation of differences in CBC ordering, that proximity to training is the key factor. Those trained more recently might have more “modern” practices.

Line 228: Not looking at patient factors strikes me as missing a big part of the story.

Author Response

Thank you for your time to review the manuscript and we appreciate for your valuable comments. Please see our response uploaded.

Reviewer 3 Report

Comments and Suggestions for Authors

1.       ABSTRACT:

2.       please explain all the abbreviations

3.       Why not mean plus minus sd in values as birth weight or gestational age that follow a normal distribution?

4.       …10 neonatologists… percentages here make no sense depict numbers instead

5.       What are the physicians characteristics ? this should be apparent in the abstract

6.       The idea under this study is very interesting but unfortunately the sample size is only 10. It is difficult to yield any clinical usefulness. It would be more interesting to distribute a questionnaire to the doctors involved to ask about the different approaches. This does not overcome the problem with the sample size but it would be a study with a different more applicant and useful orientation

Author Response

(The authors gave the same response as above.)

Round 2

Reviewer 1 Report

Comments and Suggestions for Authors

Thanks for responding to my comment. In line 98, you write "The study included all NICU patients with suspected LOS who had a blood culture obtained." However, I cannot guess from only this sentence that all patients with suspected LOS always had blood culture. Please clarify.

Author Response

Response to Reviewer 1 (round 2) Comments

1. Summary

Thank you again very much for taking the time to review this manuscript. Please find the detailed responses below and the corresponding revisions/corrections highlighted in the re-submitted files.

2. Questions for General Evaluation

Reviewer’s Evaluation

Response and Revisions

Does the introduction provide sufficient background and include all relevant references?

Yes

Are all the cited references relevant to the research?

Yes

Is the research design appropriate?

Yes

Are the methods adequately described?

Can be improved

Reviewer comments addressed, please see below

Are the results clearly presented?

Yes

Are the conclusions supported by the results?

Yes

3. Point-by-point response to Comments and Suggestions for Authors

Comment: Thanks for responding to my comment. In line 98, you write "The study included all NICU patients with suspected LOS who had a blood culture obtained." However, I cannot guess from only this sentence that all patients with suspected LOS always had blood culture. Please clarify.

Comment accepted. As a policy in our unit a blood culture is done in all suspected cases of LOS prior to initiation of antibiotics. We have clarified this in line 100

Reviewer 2 Report

Comments and Suggestions for Authors

Much improved.

A few questions remain:

Why were some of the cultures that were positive for Coagulase-negative staph considered contaminated (lines 183-184) but others were considered as positive? (lines 185-186)? This was not part of the study, but can the authors tell us about this? As I noted in my previous review, this is by far the largest group of positive cultures.

Do mention if anyone else besides the neonatologists were making decisions about LOS workups. This might include trainees (residents, fellows) and/or mid-level providers (NPs, PAs). If so, this might dilute the effects of the neonatologists’ personal characteristics on decision-making. 

What the authors describe as “academic position” is usually referred-to as “academic rank”.

Author Response

For research article

Response to Reviewer 2 (round 2) Comments

1. Summary

Thank you very much for taking the time to review this manuscript. Please find the detailed responses below and the corresponding revisions/corrections highlighted in the re-submitted files.

2. Questions for General Evaluation

Reviewer’s Evaluation

Response and Revisions

Does the introduction provide sufficient background and include all relevant references?

yes

Are all the cited references relevant to the research?

Yes

Is the research design appropriate?

Yes

Are the methods adequately described?

Yes

Are the results clearly presented?

Yes

Are the conclusions supported by the results?

Yes

3. Point-by-point response to Comments and Suggestions for Authors

Comment 1: Why were some of the cultures that were positive for Coagulase-negative staph considered contaminated (lines 183-184) but others were considered as positive? (lines 185-186)? This was not part of the study, but can the authors tell us about this? As I noted in my previous review, this is by far the largest group of positive cultures.

Comment accepted. Contaminant determination for coagulase negative staphylococcus was based on physician clinical decision to stop antibiotics in less than 5 days. We have excluded contaminants, as we did not want cases which were determined as contaminant effect our analysis of correlation of physician characteristics and a true positive blood culture. We have included this comment in line 109. And we have deleted “and physician decision to stop antibiotics in less than 5 day’s” to rephrase the paragraph.

Comment 2: Do mention if anyone else besides the neonatologists were making decisions about LOS workups. This might include trainees (residents, fellows) and/or mid-level providers (NPs, PAs). If so, this might dilute the effects of the neonatologists’ personal characteristics on decision-making. 

Comment accepted. Even though we have midlevel trainees and providers in our unit, the final decision to do a LOS evaluation is done by neonatologist. We have 24/7 in house neonatologist in our NICU. We have clarified this in line 101

Comment 3: What the authors describe as “academic position” is usually referred-to as “academic rank”.

Comment accepted. As per your suggestion we have changed “academic position” to “academic rank”. The changes were made in line 14, 90, 120, 172, 185,198, 251, table 2, table 3 and table 4

Reviewer 3 Report

Comments and Suggestions for Authors

none, my comments have been well addressed

Author Response

Thank you again for reviewing the manuscript and for your valuable input.
